# Potential for Phages in the Treatment of Bacterial Sexually Transmitted Infections

**DOI:** 10.3390/antibiotics10091030

**Published:** 2021-08-24

**Authors:** Kathryn Cater, Ryszard Międzybrodzki, Vera Morozova, Sławomir Letkiewicz, Marzanna Łusiak-Szelachowska, Justyna Rękas, Beata Weber-Dąbrowska, Andrzej Górski

**Affiliations:** 1Bacteriophage Laboratory, Hirszfeld Institute of Immunology and Experimental Therapy, Polish Academy of Sciences (HIIET PAS), 53-114 Wrocław, Poland; klcater@crimson.ua.edu (K.C.); marzanna.lusiak-szelachowska@hirszfeld.pl (M.Ł.-S.); justyna.rekas@hirszfeld.pl (J.R.); beata.weber-dabrowska@hirszfeld.pl (B.W.-D.); andrzej.gorski@hirszfeld.pl (A.G.); 2Phage Therapy Unit, Hirszfeld Institute of Immunology and Experimental Therapy, Polish Academy of Sciences (HIIET PAS), 53-114 Wrocław, Poland; 3Department of Clinical Immunology, Transplantation Institute, Medical University of Warsaw, 02-006 Warsaw, Poland; 4Laboratory of Molecular Microbiology, Institute of Chemical Biology and Fundamental Medicine, Siberian Branch of Russian Academy of Science, Ac. Lavrentyev’s Prospekt 8, 630090 Novosibirsk, Russia; morozova@niboch.nsc.ru; 5Department of Health Sciences, Jan Długosz University in Częstochowa, 42-200 Częstochowa, Poland; letkiewicz1@o2.pl; 6Infant Jesus Hospital, Medical University of Warsaw, 02-005 Warsaw, Poland

**Keywords:** antibiotic resistance, bacteriophage, endolysins, phage therapy, sexually transmitted infections

## Abstract

Bacterial sexually transmitted infections (BSTIs) are becoming increasingly significant with the approach of a post-antibiotic era. While treatment options dwindle, the transmission of many notable BSTIs, including *Neisseria gonorrhoeae*, *Chlamydia trachomatis*, and *Treponema pallidum,* continues to increase. Bacteriophage therapy has been utilized in Poland, Russia and Georgia in the treatment of bacterial illnesses, but not in the treatment of bacterial sexually transmitted infections. With the ever-increasing likelihood of antibiotic resistance prevailing and the continuous transmission of BSTIs, alternative treatments must be explored. This paper discusses the potentiality and practicality of phage therapy to treat BSTIs, including *Neisseria gonorrhoeae*, *Chlamydia trachomatis*, *Treponema pallidum*, *Streptococcus agalactiae*, *Haemophilus ducreyi*, *Calymmatobacterium granulomatis*, *Mycoplasma genitalium*, *Ureaplasma parvum*, *Ureaplasma urealyticum*, *Shigella flexneri* and *Shigella sonnei*. The challenges associated with the potential for phage in treatments vary for each bacterial sexually transmitted infection. Phage availability, bacterial structure and bacterial growth may impact the potential success of future phage treatments. Additional research is needed before BSTIs can be successfully clinically treated with phage therapy or phage-derived enzymes.

## 1. Introduction

The rise in prevalence and variety of antibiotic-resistant bacteria in recent years presents a growing problem for the medical and scientific communities and to society as a whole. A remarkable number of bacteria are resistant to at least one antibiotic, with many presenting multidrug resistance [1]. A notable few bacteria, referred to as superbugs, are resistant to most currently discovered antibiotics, and their threat to the public cannot be understated. These superbugs present treatment challenges rivaling those of a pre-antibiotic era [2]. Without preventive action, the number of people dying annually from antibiotic-resistant infections is expected to outnumber that of individuals dying annually from cancer by the year 2050 [3]. Antibiotic-resistant bacteria are rapidly disseminating throughout the world as well, markedly due to the over and improper use of antibiotics since their discovery. This erroneous usage has supported and will continue to support the repeated evolutionary development of bacterial mechanisms to evade death by antibiotic compounds.

The growing antibiotic resistance epidemic becomes especially problematic when discussing its pervasiveness among bacterial sexually transmitted infections (BSTIs). Not only has antibiotic resistance been a growing problem, but the prevalence of BSTIs, especially in the United States, has been rapidly increasing. The Centers for Disease Control and Prevention (CDC) report that *Neisseria gonorrhoeae*, *Chlamydia trachomatis* and *Treponema pallidum* have increased at unprecedented rates of 67%, 22% and 71%, respectively, from 2013 to 2017, equaling an overall rate of increase of 31%. This is an increase of 1.75 million Americans to 2.29 million Americans diagnosed with these three infections [4]. *Neisseria gonorrhoeae* is considered significantly antibiotic-resistant, with the CDC reporting it as an urgent threat to human health [5] and the World Health Organization (WHO) reporting it as a pathogen with a high priority need for new treatments [6]. *Chlamydia trachomatis* has had reported antibiotic resistance to several classes of antibiotics [7], while *T. pallidum* is still considered susceptible to its recommended antibiotic treatments [8]. In addition to these BSTIs becoming more common and harder to treat, their infections can result in lifelong repercussions, including infertility and additional life-threatening complications [4]. *Streptococcus agalactiae*, colloquially known as Group B *Streptococcus* (GBS), is recognized as a BSTI problematic during pregnancy, presenting as a growing antibiotic-resistant risk, with the CDC reporting GBS erythromycin-resistant and clindamycin-resistant infections as concerning threats [5]. Despite their low frequency, other BSTIs such as *Haemophilus ducreyi*, *Calymmatobacterium granulomatis*, *Mycoplasma genitalium*, *Ureaplasma parvum* and *Ureaplasma urealyticum* are problematic infections that can significantly impact human health: all of which, except *C. granulomatis*, have had reported antibiotic resistance [9,10,11,12,13,14,15,16,17,18]. Additionally, gastrointestinal bacterial infections, occasionally classified as sexually transmitted infections (STIs), such as *Shigella* and less often *Campylobacter* and *Salmonella*, are posing threats to human health due to increased transmission and antibiotic resistance [5,19,20,21,22,23,24]. If no new treatment methods are developed, the inability to treat these BSTIs could cause widespread disease and unprecedented consequences to the reproductive population and future generations.

Bacteriophage therapy, or phage therapy, presents hope for a solution to the burgeoning antibiotic resistance crisis. The concept of bacteriophage therapy is not novel and has been an idea since before the discovery of penicillin [25]. After the commercialization of different antibiotic compounds for therapeutic use, phage therapy became a more arduous and circuitous method to treat bacterial infections than antibiotics [26]. However, parts of Eastern Europe and the former Soviet Union continued research into bacteriophages for their use in phage therapy [25]. In subsequent years, due to the over and improper use of antibiotics, bacteria were able to develop a wide range of mechanisms to evade death at the hands of these compounds. Now, the rate at which bacteria can and are developing resistance to current antibiotics is much faster than the rate at which new antibiotics are being discovered and developed [1]. This threat drives the pursuit of effective treatment methods and has led to renewed interest in phage therapy in the West [27].

## 2. Threat of Bacterial Sexually Transmitted Infections

The potential and need for phage treatments vary significantly among the aforementioned sexually transmitted bacteria. Considerations for the potential of phage use in treatments include epidemiology, antibiotic resistance, susceptibility to current antibiotic treatments and severity of symptoms (see Table 1).

### 2.1. Epidemiology

The most frequent bacterial STI worldwide is *C. trachomatis* [7], followed by *N. gonorrhoeae* [30]. *Treponema pallidum* [4] and *S. agalactiae* [5,108] are also frequent BSTIs but with lower rates of contraction than their previously mentioned counterparts. Bacterial STIs including *M. genitalium* [9], *H. ducreyi* [13,15] and *C. granulomatis* [17] are endemic in less developed countries but tend to only have rare outbreaks in the developed world. *Ureaplasma parvum* and *U. urealyticum* [12,83] are similar to *S. agalactiae* [108] in that they are often found colonizing people asymptomatically. Unless large enough amounts of *Ureaplasma* are present to cause an infection, *U. parvum* and *U. urealyticum* are rarely problematic [12]. Exposure to the gastrointestinal bacterial *Shigella flexneri* and *Shigella sonnei* through sexual transmission has become an emerging problem [19,20,21,22,23,24,100]. This transmission is more common among men of the gay and bisexual communities; outbreaks have been observed in the United Kingdom, and transmission has been reported across Europe [23]. Gastrointestinal bacteria *Campylobacter* and *Salmonella* have also been reported to be transmitted sexually [20,109]. However, unlike *Shigella,* there have been no reported outbreaks in the developed world, and they seem to pose less of a threat as sexually transmitted infections. The frequency of infection (see Table 1) is important when considering the potential and necessity for phage application in treatment because therapeutic phage use requires an extensive amount of groundwork and research before it becomes a feasible option.

### 2.2. Antibiotic Resistance and Current Treatment Methods

Of similar importance when accessing the potential for phage use in treatment is the bacteria’s current state of antibiotic resistance and susceptibility to established treatment methods (see Table 1). With increasing antibiotic resistance and decreasing susceptibility to current antibiotic therapies, bacteria pose an expanding threat to human health. The larger the threat, the more pressing the need for alternative treatments such as phage therapy. As mentioned previously, *N. gonorrhoeae* [5], *S. agalactiae* [5], *M. genitalium* [11], *H. ducreyi* [13,15], *S. flexneri* and *S. sonnei* [5,19,24] are all considered antibiotic-resistant threats. The combination of azithromycin/ceftriaxone therapy [6] is recommended for *N. gonorrhoeae* treatment due to its pervasive antibiotic resistance. Now recognized by the CDC as a superbug, it is predicted that *N. gonorrhoeae* will soon develop resistance to all current treatment methods [30]. *Haemophilus ducreyi* and *M. genitalium* have several lines of antibiotic defense recommended, respectively: azithromycin, then ciprofloxacin or erythromycin [14]; and azithromycin, moxifloxacin, then pristinamycin or doxycycline [10]. These multiple lines of defense are required due to the ubiquitous antibiotic resistance among *H. ducreyi* and *M. genitalium*. Prior to any treatment, antibiotic susceptibility tests are strongly recommended. *Streptococcus agalactiae* [56] presents with threatening antibiotic resistance; recommended treatment includes penicillin. Cephalosporins are recommended for the treatment of *S. flexneri* and *S. sonnei* [21]. *Chlamydia trachomatis* [28], *U. parvum* and *U. urealyticum* [83] are all fairly antibiotic-resistant. The recommended treatments for these bacteria include antibiotic therapies such as azithromycin [37,83]. *Treponema pallidum* [8] and *C. granulomatis* [16,17] are not considered antibiotic-resistant threats and are still considered susceptible to penicillin [8] and azithromycin [16], respectively. The most compelling support for the use of phages in future therapeutics is to combat antibiotic-resistant threats. As we are approaching an era where current treatment options will become ineffective, other treatment methods such as phage usage will be necessary. Several BSTIs pose significant antibiotic-resistant threats, making the need for alternative treatments for these BSTIs more pressing than for their counterparts that remain susceptible to antibiotic treatments.

### 2.3. Symptoms of Infection

Another important factor to examine when considering the need for phage therapy is the severity of symptoms associated with the pathogen. If, and when, antibiotic therapies become ineffective, what will be the impact on human health from the BSTI in consideration? Bacterial sexually transmitted infections are often not regarded as deadly due to the ability to treat and remove these infections. In contrast, virally sexually transmitted infections may be treated symptomatically while the infection remains. The ‘curability’ of BSTIs decreases concerns, however, when left untreated. Serious symptoms can develop, resulting in life-threatening complications (see Table 1). Growing antibiotic resistance and the inability to treat these BSTIs is rapidly becoming a prospective reality. Posing issues for diagnosis, treatment and transmission of BSTIs, *N. gonorrhoeae* [110], *C. trachomatis* [7] and *T. pallidum* [48] can present asymptomatically, leading people to not seek treatment until symptoms worsen. *Neisseria gonorrhoeae* and *C. trachomatis* deterioration can include inflammation of the urogenital tract in both men and women, including pelvic inflammatory disease, urethritis, proctitis, cervicitis, epididymitis, prostatitis, salpingitis and endometritis [7,31,110]. *Neisseria gonorrhoeae* may cause a disseminated gonococcal infection, a life-threatening sepsis that can occur if left untreated for an extended period of time [31,110]. *Treponema pallidum* presents in several stages, including primary syphilis, secondary syphilis, latent syphilis and tertiary syphilis; these stages represent a worsening of symptoms as the *T. pallidum* infection progresses, excluding the asymptomatic latent stage [48]. Primary, secondary and latent syphilis do not pose life-threatening issues, often presenting with a painless localized lesion, a papular rash and a lack of symptoms, respectively. Tertiary syphilis can lead to cardiovascular, neurological and gummatous syphilis, all of which have life-threatening implications. In addition, these BSTIs may also become especially problematic during pregnancy, not only for pregnant women, but for the fetus or newborn. *Neisseria gonorrhoeae* [110,111], *C. trachomatis* [7], *T. pallidum* [48], *S. agalactiae* [55], *U. parvum* and *U. urealyticum* [12] can all cause issues during pregnancy. *Neisseria gonorrhoeae*, *C. trachomatis, U. parvum* and *U. urealyticum* can all cause premature delivery. *Treponema pallidum* can cause stillbirth or fetal infection up to 4 years after infection in the mother [48]. *Streptococcus agalactiae* is the leading cause of neonatal sepsis and meningitis [55], and *C. trachomatis* can cause chorioamnionitis, neonatal conjunctivitis and neonatal pneumonia [7]. *Neisseria gonorrhoeae* can cause pelvic sepsis in the mother and gonococcal ophthalmia neonatorum and systemic neonatal infections in the newborn [111]. Additionally, *N. gonorrhoeae* [110], *C. trachomatis* [7], *U. parvum*, *U. urealyticum* [12] and *M. genitalium* [9] can also lead to infertility in women, with some reports suggesting *M. genitalium* leading to decreased sperm count in men as well [9]. Infection with some BSTIs can increase the risk of co-infection with other sexually transmitted infections. *Ureaplasma parvum* and *U. urealyticum* have been related to *C. trachomatis* co-infection [83], while *C. trachomatis* is associated with HIV infection and cervical cancer [7]. *Ureaplasma parvum* and *U. urealyticum* are also associated with HIV co-infection and have been reported to be related to certain renal infections [83]. If untreated, *C. granulomatis*’s lymphadenopathy can result in genital mutilation so extensive that surgery may be required [17]. *Chlamydia trachomatis* can also cause lymphadenopathy known as lymphogranuloma venereum (LGV), and if untreated, it can lead to secondary infections, fistulas and ulcers [38]. *Haemophilus ducreyi*, the causative agent of chancroid, causes ulcers of the skin and genitals [13], while *S. sonnei* and *S. flexneri* cause intestinal diarrheal disease shigellosis [21]. Many BSTIs, especially when left untreated, result in serious complications that can become life-threatening, while others remain less severe. Considering frequency, antibiotic resistance, susceptibility to current antibiotic treatments and severity of symptoms is paramount to identifying the demand for phage therapy for specific BSTIs.

## 3. Challenges Associated with Application of Phage Therapy in Bacterial Sexually Transmitted Infections

The challenges associated with phage therapy must also be considered when determining the potential of phage therapy for each BSTI. Some important considerations for the potential for future phage treatments include phage availability, as well as the bacterial structure and the bacteria’s growth nature (see Table 1). Issues such as the mode of application, purification, stability, safety and any regulatory concerns that may pose problems at later stages of phage therapeutic development must also be considered.

### 3.1. Phage Availability

To date, *C. granulomatis* [18], *U. parvum* and *U. urealyticum* do not have any isolated phages reported [40]. There is debate on whether to classify *Calymmatobacterium granulomatis* as *Klebsiella granulomatis*. There have been many *Klebsiella* phages isolated, characterized and even used in clinical phage therapy [17]. However, there is no evidence to support that any of these *Klebsiella* phages may infect *C. granulomatis*. In addition, since there is debate surrounding the genetics of the classification of these bacteria as *Calymmatobacterium* or *Klebsiella*, it is exceedingly unlikely that these *Klebsiella* phages would have a host range including *C. granulomatis* [76]. The genera *Treponema* [51,53,54], *Mycoplasma* [77,78,80,81,82,112] and *Haemophilus* [93,94,95] all have isolated phages but not for the causative organism of the BSTI. However, there is potential for phages to exist for the species of interest if they can exist within their genera [54], which also poses relative potential for *Ureaplasma parvum* and *Ureaplasma urealyticum* as they are of the *Mycoplasmataceae* family, including both the *Mycoplasma* and *Ureaplasma* genera. Recently, ΦCPG1 phage’s ability to infect not only *Chalmydia caviae* but, more importantly, *Chalmydia trachomatis,* has been confirmed [41,46]. Within the *N. gonorrhoeae* genome, several prophages have been identified [34,35]. *Neisseria gonorrhoeae* [36] and *S. agalactiae* [59,61,113] have isolated lysogenic phages. The isolated *N. gonorrhoeae* [36] lysogenic phage is from an activated prophage within its genome, while *S. agalactiae* [59,61,62,113] phages have been isolated environmentally. *Shigella flexneri* and *S. sonnei* have virulent phages [102,103,104,105,106], some of which are being used clinically to treat shigellosis [25,98]. However, these treatments have not been considered for use in the treatment of STIs, as historically, shigellosis is obtained through the traditional fecal–oral route. This clinical evidence provides hope that *S. flexneri* and *S. sonnei* infections obtained through sexual transmission may also be treated in a similar manner with phage therapy.

Genetic modification of existing phages may significantly help to produce novel bacteriophages with unique therapeutic properties, as reviewed in detail by experts in [114]. Both traditional and continually developing modern technologies such as BRED (bacteriophage recombineering of electroporated DNA) or CRISPR-Cas system enable to obtain phages of extended host range and efficacy, improved antibiofilm activity, stability (resistance to the gastric juice barrier) and even pharmacokinetics (long-circulation in the blood stream). One of the most important advantages of phage engineering is the possibility to obtain obligatory lytic phages devoid of virulence genes by the removal of appropriate genomic modules from temperate phages. M13, a filamentous *E. coli* phage, was modified to stably express an integrin-binding peptide and a conserved polymorphic membrane protein D from *C. trachomatis* [115]. The first one is responsible for phage endocytosis, whereas the second one interferes with the propagation of the bacterium and is probably responsible for the phage homing to the inclusions. In this form, it was able to inhibit *C. trachomatis* infection both in HeLa cells and endocervical cells (isolated from primary human endocervical tissue), presenting a potential to control chlamydial infections.

### 3.2. Potential for Enzymatic Phage Therapy

Lysogenic phage genomes provide valuable information that can be utilized in enzymatic phage therapies. However, because they are able to incorporate their genetic material into the host genome, causing unpredictable consequences [116], they are not useful for whole-phage therapy. Phage action is widely different from antibiotic mechanisms. Unlike antibiotics, which are molecular compounds, bacteriophages have life cycles. During the phage life cycle, degradative enzymes (see Figure 1) known as endolysins (supported by small pore-forming transmembrane peptides called holins) and depolymerases are able to degrade parts of the bacterial cell wall, membrane or capsule [117,118,119,120]. Studies show that the extracellular application of some of these enzymes can have antibacterial effects: slowing cell growth, degrading cell walls or resulting in cell death depending on the efficacy of the enzyme and its specificity to the host [118,120]. These are especially effective on Gram-positive bacteria, as their bacterial cell wall is exposed, compared to Gram-negative, where the enzymes must face two cell membranes and a cell wall (see Figure 1A,B). Due to the resistance of the peptidoglycan cell wall in Gram-positive bacteria, degradation of this layer will result in cell lysis, while in Gram-negative bacteria, cell membrane penetration is often needed. Little is known about how phage enzymes function within a *Mycoplasmataceae* bacterium and how these enzymes could be utilized for therapeutic means (see Figure 1C). Bacteriophages exist for *Mycoplasma* and are able to perpetuate their life cycle within them, suggesting that phage enzymes for lysis may exist [78,80,81,82,113]. However, there are no experimental data supporting this hypothesis, only theoretical notions and comparisons with Gram-negative and Gram-positive phages [76]. Occasionally, both Gram-positive and Gram-negative bacteria may be encapsulated, creating an additional outer layer to be degraded (see Figure 1D,E). Complicating treatments further are biofilms, which may be observed in a variety of bacteria. Biofilms are notoriously resistant to antibiotics; however, certain phages possess the ability to effectively degrade biofilms where antibiotics fail (see Figure 1F,G) [121,122]. Interestingly, CHAP_k_ lysine—a derivative of native lysin of *S. aureus*—presented strong activity against *S. agalactiae* biofilm even more potent than vancomycin [67]. Currently, work is being performed to biologically engineer hybrid phage enzymes for both Gram-positive and Gram-negative bacteria. Certain engineered enzymes (such as Artilysin^®^s, which use lipopolysaccharide-destabilizing peptides fused to endolysin or Innolysins, which combine endolysin and phage receptor binding proteins) have been shown to have success, supporting apparent outer cell membrane targeting [123,124]. Recombinant enzymes with active sites derived from different bacterial species have been successfully shown to retain the lytic ability of both parent enzymes [125]. It is important to note that endogenous phage enzymes may have multiple active sites, and therefore multiple functional activities. For example, lysins may possess peptidoglycan-cleaving activity, resulting in a single enzyme that may behave as an endopeptidase and a glucosidase that probably enhances the outcome of their lytic activity [63]. Targeting intracellular bacteria by phage-derived enzybiotics as their intrinsic activity, as observed for PlyC, which kills *S. pyogens* inside epithelial cells, is rarely reported [126]. However, there are attempts proposed as a fusion of endolysin with cell-penetrating peptides to solve this problem [122,127].

Phages, as well as their derivative enzymes, provide potential means for the development of treatment methods for future phage therapies [128]. Evidence indicates that enzymes from the isolated *Streptococcus* and *Chlamydia* phages can target *S. agalactiae* and *C. trachomatis* to impact their growth. Phage enzymes targeting *C. trachomatis* have displayed success in vitro [42,43,44,45]. In addition to genetically engineered recombinants [125], phage enzymes targeting *S. agalactiae* have been successful in vitro [60,63,64,65,72] and in vivo [68,73,74]. For *C. trachomatis*, reports indicate that φCPG1 phage capsid protein Vp1 has disruptive effects on its growth in vitro [42,43,44,45]. φCPG1 itself has shown a delayed appearance of *Chlamydia caviae* in a Guinea pig animal model and a decreased pathological response [46]. While for *S. agalactiae,* evidence indicates successful in vitro activity against *S. agalactiae* from phage lysins derived from *S. agalactiae* phages B30 [63,64], λSA2 [65] and λSA1 [72] *Staphylococcus aureus* and *Streptococcus agalactiae* recombinant enzyme derived from B30 endolysin and lysostaphin [125] as well as previously mentioned CHAPk [67]. In addition, in vivo analysis demonstrates the use of lysin PlyGBS derived from *S. agalactiae* phage NCTC 11261 in the reduction of GBS colonizing the vagina and oropharynx in mice [68]. Chimeric ClyV lysin (obtained by fusion of the enzymatically active domain from PlyGBS lysin and the cell wall binding domain from PlyV12 lysin) presented improved bactericidal activity in vitro and protected mice from lethal infection caused by intraperitoneal injection of *S. agalactaie* with a good safety profile [69]. *Streptococcus agalactiae* holds a great deal of potential for use in enzymatic phage therapy for many reasons, as it is Gram-positive, it only has lysogenic bacteriophages characterized, its topical application of phage enzyme preparations has reasonable support and GBS only becomes problematic when a woman is pregnant. Species-targeted selection would still occur with enzymes, so phage enzymes have the additional benefits over antibiotic prophylaxis by leaving the mother’s original microflora intact for delivery. This is especially advantageous as the vaginal microflora has been shown to be beneficial for immune system development in the neonate [129]. Additionally, phage enzyme treatments do not need to be limited to the mother but could also be used as an additional decontamination treatment for the newborn after birth [68].

Some approaches, such as screening of uncultured viral genomes, could help to discover new endolysin genes for those BSTI pathogens and their phages for which culturing is really challenging [130]. Although bacterial vaginosis, which is caused by an imbalance in the vaginal flora and an overgrowth of *Gardnerella,* is not recognized as a BSTI, one of the recent reports showed that the genus-specific effect of the PM-477 endolysin was shown not only in suspension but also ex vivo on biofilms of *Gardnerella* on exfoliated vaginal epithelial cells obtained from patients with bacterial vaginosis [131]. This is a good example that it is possible to overcome some challenges and to produce an engineered phage endolysin even despite the lack of isolated phages with significantly improved protein’s shelf-life and antibacterial activity.

### 3.3. Current Clinical Possibilities and Challenges

To date, no current clinical phage therapy has been reported in the treatment of sexually transmitted infections, both within Georgian literature published in English or among Russian literature. However, phage therapy of the urogenital tract has been performed as well as the previously mentioned phage therapy of *S. flexneri* and *S. sonnei* as agents of dysentery. In the beginnings of phage therapy, Felix d’Herelle, one of the discoverers of bacteriophages, utilized phages to treat dysentery. Since that time, *S. flexneri* and *S. sonnei* treatments and prophylaxis have been occurring with bacteriophages throughout Eastern Europe and the former Soviet Union [25,107]. The current studies of phage application in the treatment of vaginal infections with bacterial agents [25,132] provide support that with appropriate phages, the topical application process may provide successful results that are transferrable to BSTI treatment. Additionally, transferrable to the possible treatment of BSTIs, are current clinical trials that provide information on the importance of phage stability, the importance of regulations surrounding how phages can be used in treatment and how phage properties may impact the success of a study. The PhagoBurn study was the first double-blind, randomized, clinical trial of phage therapy that offers some insight into future challenges in that fight [133]. Although the main cause of the fail of this trial was pure stability of the applied phage cocktail, current studies propose many effective solutions to solve such problems [134]. They include the use of different stabilizers according to the required form of phage application, as well as lyophilization and encapsulation [134,135,136,137]. Another recently completed randomized, double-blind clinical trial of intravesical phage therapy in treating urinary tract infections does not suggest the efficacy of phage instillation to treat bacterial infections [138]. Therefore, a formal confirmation of the therapeutic value of phage therapy in urogenital tract infections according to standards of evidence-based medicine is still required. Hopefully, a regulatory crisis caused by difficulties in the qualification of the nature of phage preparations according to pharmaceutical law and interpretation of its different aspects should be filled before their registration, has already been solved. Leading regulatory authorities such as the US Food and Drug Administration and the European Medicines Agency elaborated clear basic hints for sponsors interested in clinical trials as well as for therapeutic expanded access to therapeutic phage products [139,140].

As already pointed out, there is a number of challenges that could affect the possible successful use of phage therapy in BSTIs. This also includes a virtual lack of knowledge on the pharmacokinetics of phages administered in humans, optimal routes of their administration, phage penetration in the urogenital tract as well as phage immunogenicity and its relation to the outcome of the therapy [141]. Notably, sera of patients infected with *C. trachomatis* may contain anti-phage antibodies [43]. Furthermore, we recently demonstrated that intravesical phage administration does not induce significant neutralizing antibody responses to phages [142]. On the other hand, intravenous phage administration may cause robust antibody responses with concurrent limited therapeutic efficacy [143]. Therefore, the mode of phage administration in the treatment of BSTI and its relation to the outcome of the therapy should also be carefully considered. Interestingly, our observations suggest that although oral or topical phage administration (which would be preferred in the treatment of many BSTIs) may induce an immune response to phages in some patients; it does not absolutely affect the final clinical outcome [144,145].

### 3.4. Challenges Posed by Bacterial Properties and Conditions

In addition to bacteriophages and clinical trials, a bacterium’s nature, such as its cell wall structure, capsule presence, growth conditions and cultivability, all play a role in its potential for phage therapy. Of all the BSTIs mentioned, *S. agalactiae* is the only BSTI with a Gram-positive cell wall and one that presents 10 different capsule serotypes (Ia, Ib, II-XI) [58]. It grows easily under laboratory conditions and only has lysogenic bacteriophages isolated. These attributes are likely why there is extensive knowledge about its potential for use in phage enzymatic therapy. Gram-negative bacteria include *T. pallidum* and *C. trachomatis,* both of which are without capsules, while *C. trachomatis* [146] is also an intracellular parasite. In a laboratory setting, *T. pallidum* [50] is dependent on a model organism for reproduction, while *C. trachomatis* is propagated through cell line cultures. Since these are both challenging organisms to culture and grow, this supports the limited evidence available on *T. pallidum* and *C. trachomatis* bacteriophages. *Shigella sonnei* and *S. flexneri* are Gram-negative as well, and *S. sonnei* has a capsule, while *S. flexneri* does not [101]. *Shigella sonnei* and *S. flexneri* grow robustly in a lab, making them useful organisms for experimentation. Since the inception of bacteriophage therapy, these two organisms have been at the frontlines. However, as they have just recently begun to pose issues as STIs, with specific hotspots in the developed world, there is little evidence in their connection between phage therapy and sexual transmission. Despite this, there is abundant research with *S. sonnei* and *S. flexneri* phage therapy as gastrointestinal bacteria, and it is likely that this will be beneficial to its treatment as an STI. *N. gonorrhoeae* and *H. ducreyi* are both Gram-negative encapsulated bacteria; however, *H. ducreyi*’s capsule is not classical but rather a loose capsular structure [89,90]. *Calymmatobacterium granulomatis* is a Gram-negative encapsulated intracellular parasite that can be observed outside its host cell [17,18]. A few options were proposed to solve the problem of targeting obligatory intracellular bacteria and those ones that may hide within host cells (as, for example, discussed here: *S. agalactiae*) [147]. One of them is genetic engineering, as it was described earlier in this paper for modified phage M13 infecting *C. trachomatis*. [115]. The other may be the formulation of liposomes as active phage carriers that could be internalized by eukaryotic cells or the use of phage composite with inorganic nanoparticles (for example, hydroxyapatite) that stimulate phage uptake by cells [148]. *Neisseria gonorrhoeae* [33], *H. ducreyi* [13,91] and *C. granulomatis* [17,18] all grow best under microaerophilic conditions with high levels of CO_2_. As *H. ducreyi* [13,91] and *C. granulomatis* [17,18] are more rare infections that require microaerophilic growth conditions, little evidence for either of their bacteriophages exists. *Neisseria gonorrhoeae* [33] poses much more of a threat. However, because of its challenging microaerophilic growth conditions, there is little renewed interest in investigating phage therapeutics as alternatives to antibiotics. *Mycoplasma genitalium* [76], *U. parvum* and *U. urealyticum* [83] have no cell wall, and *M. genitalium* [10,76] has no capsule. *Ureaplasma urealyticum* has been experimentally shown to possess a capsule, and a capsule is hypothesized to exist in *U. parvum* [84] as well. *Haemophilus ducreyi* [89,91], *M. genitalium* [10], *U. parvum* and *U. urealyticum* [83] all have supplemental media requirements for growth (see Table 1), while *U. parvum* and *U. urealyticum* requires a pH indicator to signify growth, as turbidity does not appear [83]. The challenging growth conditions presented by *M. genitalium* [10], *U. parvum* and *U. urealyticum* [83], as well as their rarity as BSTIs, could account for the little bacteriophage research for any of these bacteria. In addition, as *Mycoplasmataceae,* these three BSTIs are more difficult to work with in general. *Treponema pallidum* [50], *H. ducreyi* [13,91], *C. granulomatis* [17,18] and *M. genitalium* [10] all grow slowly. This slow growth can place a strain on experimentation and make these organisms more challenging to research. The lack of bacteriophage research into these four hosts is either that they are a rarer BSTI or that they do not pose a significant enough antibiotic-resistant threat. Despite this, in 1947, Zaeva reports the use of anaerobic bacteriophages against *Clostridium* species in animal models [107]. It would suggest that regardless of a bacterium’s challenging growth conditions, successful isolation and therapy may be possible, thus providing hope for future phage therapy on many of the fastidious BSTIs.

Taking under consideration the unique culture requirements of some above-mentioned bacteria, it seems that there is still an open field to improve current techniques to isolate phages for BSTI treatment. Classical methods usually test environmental samples (believed to contain any phages) on solid bacterial lawns to produce any visible plaques that result from phage multiplication within the used host [149]. The first challenge is “enrichment” or the concentration of putative phages in a liquid sample using different approaches and its preservation [149,150]. The second one is the adjustment of incubation conditions and media. The next problem may be the detection of phages that produce obscure plaques or stop growing in liquid cultures before visible lysis occurs. Hopefully, there are proposed methods that may solve this problem, such as the use of sublethal doses of antibiotics [151] or the use of special gels which enable large phages to diffuse more to facilitate visualization of their plaques or are dedicated for those phages not propagating in traditional agar gels [152]. Isolation and propagation of phages of obligatory intracellular pathogens are much more complicated. They require incubation of the phages and bacteria in the presence of eukaryotic cells in which the process of phage multiplication takes place [153]. Next, the phage is released from the cells by a repeated freezing and thawing cycle. This technique may not be efficient enough to produce phages on an industrial scale. In such cases, the use of adopted phages or phage lysins may be an alternative option.

## 4. Conclusions

Each BSTI presents its own challenges when it comes to its potential for use in phage therapy. Further studies using animal models and clinical trials are necessary to assess the value of phage therapy in their treatment. Therefore, phage therapy treatments in a primary care facility are a long way from realization, especially for BSTIs. With the increasing frequency and antibiotic resistance of some of these BSTIs, most notably *N. gonorrhoeae*, humans are entering a dangerous post-antibiotic era. Despite the extensive work required for successful clinical phage therapy and the challenges associated with each individual BSTI, phage therapy provides possibilities in the fight against antibiotic resistance. Without alternative treatments, human health will suffer.

## 5. Expert Opinion

Several BSTIs present more promising opportunities for phage therapy, while others will not be very successful with current methods of phage isolation. *Streptococcus agalactiae*, *C. trachomatis*, *S. sonnei* and *S. flexneri* are the closest of the BSTIs to begin to be treated with phages and their derivatives. Successful results have been achieved with whole phages and their enzymatic counterparts in vitro, in vivo and clinically. *Mycoplasma genitalium*, *U. parvum* and *U. urealyticum* pose challenges for phage therapeutics. With *Mycoplasmataceae,* little is known in general about their phage biology. However, due to their basic cell membrane and lack of a cell wall, they may present parallels with Gram-negative bacteria in the obstacles they present with phage therapeutics [76]. *Haemophilus ducreyi* and *C. granulomatis* present with low rates of infection, similar to that of *Mycoplasma* and *Ureaplasma*, making their need for phage therapy less pressing. *Treponema pallidum* holds little information about its possible phage biology. Despite its high rates of infection, it remains antibiotic susceptible, thus decreasing the need for alternative treatments. *Neisseria gonorrhoeae* poses the largest threat of the discussed BSTIs, with its high antibiotic resistance, transmission and infection rates. In addition, despite the pressing need, there is limited knowledge on its phage biology. For all of these BSTIs, current antibiotic resistance rates pose challenges for current therapies, and new methods of treatment must be developed.

The inherent properties of phages in comparison with antibiotics make them very different from antibiotics in clinical use. In the past, broad-spectrum antibiotics provided ease of treatment; however, increasing antibiotic resistance prevents their use. In comparison with antibiotics, phages have narrower, more species-specific host ranges and additional complexity, such as their requirements for a replication life cycle. Tailored phage therapy requires more work than prescribing a broad-spectrum antibiotic for a suspected bacterial infection. The process begins with the identification of the pathogen, followed by subsequent susceptibility tests of available phages that require culturing of the pathogen. This, in itself, is known to be challenging for several BSTIs mentioned. Phage cocktails can be utilized without such issues; however, they face their own challenges [154]. Cocktails require a carefully selected phage cohort to cover a broad spectrum, controlled stability and activity, as well as high purity levels. Stability, activity and purity are important with all phage preparations [133]. As expected, these cocktails may face similar challenges to broad-spectrum antibiotics, but to much lower degrees. They still have a tailored host range, and increasing the phage number in the cocktail can provide additional selective pressure to prevent resistance from developing [154]. Specific laws, regulations and additional requirements must be met as phages are not recognized as pharmaceuticals in the same capacity as antibiotics. Additional experiments in animal models followed by successful clinical trials in humans must occur [155]. In addition, all of this work must be completed on top of a baseline of already discovered and characterized bacteriophages, which for many of the BSTIs mentioned are not present. This extensive amount of work could delay the realization of phage therapy for the treatment of BSTIs for several years to come. Despite all these challenges, as recently pointed out, clinical phage therapy seems to be a promising and safe strategy for combating antimicrobial resistance [156].

## 6. Article Highlights

The rates of certain BSTIs are increasing at unprecedented rates, while antibiotic resistance prevalence continues to grow. Bacteriophage therapy holds hope as an alternative to failing antibiotic therapies.
*S. flexneri* and *S. sonnei* phages have been utilized historically in phage therapy, and their transition to treatment in BSTIs may become possible.*M. genitalium*, *H. ducreyi*, *U. parvum* and *U. urealyticum* are rarer BSTIs that pose antibiotic-resistant threats and present challenges associated with culturing. Despite their rarity, phage therapy should eventually be investigated as an alternative treatment method.Due to their antibiotic susceptibility, *C. granulomatis* and *T. pallidum* do not necessarily require future phage therapy, although this kind of treatment poses potential.*S. agalactiae* and *C. trachomatis* pose more potential, as in vitro and in vivo studies have already shown success with phage enzyme treatment.*N. gonorrhoeae* with prophages and lysogenic bacteriophages identified holds potential for therapy using phage-derived enzymes, and due to its already high and still increasing antibiotic resistance rates, this should be investigated as an alternative treatment method.

## Figures and Tables

**Figure 1 antibiotics-10-01030-f001:**
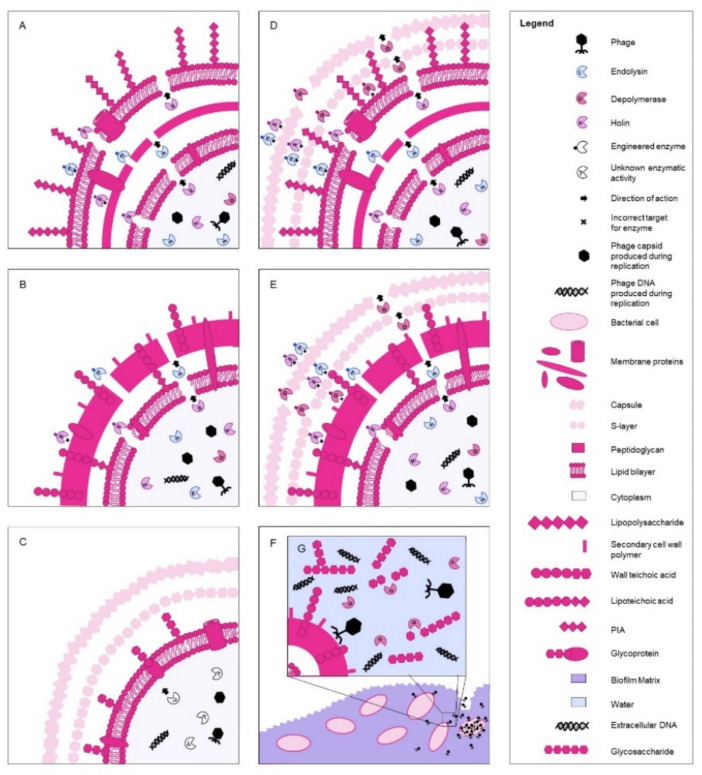
Phage/enzymatic action on a variety of cell types (see legend for names of components). Arrows indicate the direction of action of the enzymes during a natural phage infection cycle. Engineered enzymes are shown on the outside of the cells, demonstrating their ability or inability to effectively lyse the cell when applied topically. These schematics represent a general portrayal of phage/enzymatic action on a variety of cell types when each individual interaction with a certain phage, enzyme and cell type can present a more complex exchange. (**A**) Holin and endolysin activity on a Gram-negative bacterial cell. (**B**) Holin and endolysin activity on a Gram-positive bacterial cell. (**C**) Unknown enzyme activity on a mycoplasma. (**D**) Depolymerase, holin and endolysin activity on a Gram-negative bacterial cell with a capsule and s-layer shown. (**E**) Depolymerase, holin and endolysin activity on a Gram-positive bacterial cell with a capsule and s-layer shown. (**F**) Phage activity penetrating a biofilm to infect, replicate within and lyse a bacterial cell. (**G**) Shown are components of the biofilm, as well as depolymerase activity on the biofilm matrix to perpetuate phage infection into a Gram-positive bacteria cell.

**Table 1 antibiotics-10-01030-t001:** Relevant information for consideration of usefulness of phage therapy. Included is information on antibiotic resistance, relative frequency, symptoms and current treatment methods of the bacterial STIs mentioned. In addition, the phenotypic characteristics of the bacterial cell including the cell wall presence and capsule presence, as well as the bacteria’s ideal or required growth conditions. Finally, the phages available for each bacterial STI as well as relevant phage research are included.

Bacteria	Antibiotic Resistance	Frequency	Symptoms	Treatment	Cell Surface	Culture Conditions	Phages Available
*Neisseria gonorrhoeae*	Considered an urgent threat by the CDC [5], cephalosporins are the last class of antibiotics antibiotic-resistant *N. gonorrhoeae* remain susceptible to for treatment [28]; high resistance (23%) to tetracycline reported as well as cases of reduced susceptibility to azithromycin and/or cephalosporins [5]	Estimated 78 million per year worldwide [29], second most common bacterial STI [30]	Usually asymptomatic, may include: urethritis, proctitis, pharyngitis, cervicitis, chronic pelvic pain, pelvic inflammatory disease (PID), infertility, ectopic pregnancy, epididymitis, prostatitis, urethral stricture and disseminated gonococcal infection [5,29]. During pregnancy can result in premature delivery, gonococcal ophthalmia neonatorum and systemic neonatal infections in the newborn and pelvic sepsis in the mother [31]	Combination ceftriaxone plus azithromycin or doxycycline [5,28]	Gram-negative with capsule [32]	Requires 5% CO_2_ [33]	Nine identified prophages present in *N. gonorrhoeae* genome [34,35], including lysogenic filamentous phage Ngoφ6 with demonstrated activity against a variety of Gram-negative bacteria [36]
*Chlamydia trachomatis*	Some reports of macrolide and tetracycline resistance [37], treatment of lymphogranuloma venereum or LGV, with antibiotics is still considered successful [38]	Estimated 101 million per year worldwide, most common bacterial STI [7]; LGV is rare in the developed world, but outbreaks occur sporadically [38]	Asymptomatic in 75% of women and 50% of males. Symptoms include urethritis, epididymitis, proctitis, cervicitis, salpingitis, endometritis, pelvic inflammatory disease in 20% of women, infertility and ectopic pregnancy [7]. Infection can increase the risk of HIV transmission and cervical cancer [7]. Can cause preterm delivery, chorioamnionitis, neonatal conjunctivitis and neonatal pneumonia [7]. LGV causes inguinal or femoral lymphadenopathy; untreated LGV can result in secondary infections as well as genital and rectal proctocolitis, ulcers and fistulas [28,38]	Azithromycin, Doxycycline or Erythromycin [28,37]	Gram-negative intracellular parasite without capsule [28]	Propagated within cell cultures [7]	To date, five Chlamydiaphages described (Chp1, Chp2, Chp3, Chp4, ΦCPG1, ΦCPAR39) [39]; these bacteriophages have circular single-stranded DNA genomes, an estimated 6 kbp in length. *Chlamydiamicrovirus* have icosahedral, non-enveloped capsids with diameters of approximately 30 nm. They are group II bacteriophages from the family *Microviridae* and the subfamily *Gokushovirinae* [40]. Reports indicate that ΦCPG1 capsid protein VP1, as well as ΦCPG1 phage itself, has disruptive effects on the growth of *C. trachomatis* in vitro [41,42,43,44,45]. In vivo φCPG1 delays appearance of *Chlamydia caviae* and decreases pathological response in a Guinea pig animal model [46].
*Treponema pallidum* subspecies *pallidum*	Reported macrolide resistance in the US, Europe, China and Australia [8]	Estimated 12 million per year worldwide [47], from 2013 to 2017 in the US, number of cases increased by 76 percent [4]	Primary syphilis typically presents with a painless localized lesion healing on its own. Secondary syphilis often presents with a papular rash. Latent syphilis can occur for any amount of time in which a person is infected with syphilis but is asymptomatic. Tertiary syphilis occurs in 35% of people with latent syphilis, resulting in life-threatening conditions including cardiovascular syphilis, gummatous syphilis and neurosyphilis. If a baby is born when the mother is infected with *T. pallidum*, or up to 4 years after, it can cause infection in the fetus (1/3) or stillbirth (1/3); 1/3 the babies are unaffected [48]	Long-acting penicillin [8]	Weakly Gram-negative without capsule [49]	Propagated within rabbits through intratesticular, intradermal, intravenous or intracisternal inoculation; slow doubling time (30–33 h) [50]	Phages observed for the *Treponema* genus (*T. phagedenis, T. hyodysenteriae* [51,52]) but none characterized for *T. pallidum*, *T. denticola* phage described (lysogenic φtd1) [53]; phages from the spirochete phylum have been isolated and described with the majority being *Myoviridae* [54]
*Streptococcus agalactiae* (GBS)	Considered a concerning threat by CDC, clindamycin resistance prevalent, some erythromycin, azithromycin and vancomycin resistance reported [5]	Considered a part of normal flora for 10–30% of women [55], the CDC estimates 27,000 severe cases of GBS infections in the US, with 49% (~13,230) being erythromycin-resistant and with 28% (~7560) being clindamycin-resistant [5]; a majority of infants colonized with GBS do not develop a GBS infection; about 60% of cases of early-onset GBS infection occur in neonates born to patients with negative GBS culture at 35–37 weeks [56]	Leading cause of neonatal sepsis and meningitis [55]; asymptomatic in colonized women [57]	Intravenous penicillin G (during labor); ampicillin or vancomycin may be substituted [56]	Gram-positive, with ten different capsular types (Ia, Ib, II-XI) [58]	Growth observed with normal laboratory conditions (37 °C and enriched media) [59,60]	Temperate phages have been isolated and characterized for *S. agalactiae* [59,61,62]; phage lysins have successfully shown activity in vitro (lysins from *S. agalactiae* phages B30 [63,64], λSA2 [65] and λSA1 [66], as well as CHAP_k_ lysin derived from *S. aureus* [67]) and in vivo (PlyGBS from phage NCTC11261 [68] and chimeric ClyV [69]). There can be a wide host range with streptococcal phages and phage enzymes [70,71] and streptococcal lysins from other species have demonstrated successful activity in vitro (*S. dysgalactiae* subsp. *equisimilis* SK1249 prophage lysin PlySK1249 [60], *S. equisimilis* subsp. *equi* lysin PlyC [72]) and in vivo (C1 phage lysin [73]), while whole phages such as *S. pneumoniae* lytic phage PaI have had successful in vivo activity [74]
*Mycoplasma genitalium*	Reported resistance to tetracyclines, quinolones (moxifloxacin) and macrolides (azithromycin) [9], with resistance increasing at a rapid rate [10]; later-generation antibiotics are last line of defense [10]	Highly variable rates of prevalence geographically (ranging from 0% to 47.5%); estimated rates of 2.0% in low-risk groups and 7.3% in high-risk groups [9]; cause of 10–35% of non-chlamydial non-gonococcal urethritis in men [11]	Frequently presents asymptomatically; however, can cause vaginal discharge, dysuria, urethritis, cervicitis, pelvic inflammatory disease (PID), abdominal pain and dyspareunia [11]; linked to female [75] and male infertility (decreased sperm count) [9]	First line: azithromycin or josamycin; second line: moxifloxacin; third line: doxycycline or pristinamycin [10]	No cell wall (*Mycoplasma*) [76], no capsule demonstrated on *M. genitalium* [10,76]	Requires supplemental media (recommended SP4 media); fastidious and slow-growing (may take several weeks or months to grow in culture) [10]	No bacteriophages reported for *M. genitalium*. Two sequences reported for other mycoplasma viruses (*M. pulmonis* virus P1 [77] and *M. arthritidis* virus MAV1 [78]); additional mycoplasma viruses have been reported without sequence information (*M. hyorhinis* virus Hr 1 [78], *M. bovirhinis* virus Br1 [79], *M. fermentans* prophage φMFV1 [80] and mycoplasma viruses L1, L2, L3, BN1 and L172 [81,82])
*Ureaplasma parvum* and *Ureaplasma urealyticum*	Reported resistance to macrolides, tetracyclines and fluoroquinolones [83]	High prevalence of ureaplasma colonization in the healthy population (70–80%); however, infection can be dangerous. More often found in symptomatic women than asymptomatic women, *U. parvum* more frequently isolated than *U. urealyticum* [12]	Can cause renal infections as well as adverse outcomes in pregnancy such as premature labor, miscarriage or stillbirth. Additionally, may cause infertility if left untreated, may present asymptomatically or with severe symptoms in urogenital infections in women, while in men, typical presents with urethritis. *Ureaplasma parvum* and *Ureaplasma urealyticum* considered pathogenic isolates [12]	Azithromycin, doxycycline or erythromycin [83]	No cell wall (*Mycoplasma*) [83]; capsule experimentally shown to exist in *U. urealyticum* and hypothesized to exist in *U. parvum* [84]	Requires serum, growth factors and metabolic substrate (recommended SP4 media); grows without turbidity (pH indicator required for growth detection) [83]	No characterized bacteriophages reported for *U. parvum* or *U. urealyticum* [40]
*Haemophilus ducreyi*	Reported resistance to ampicillin, tetracyclines, sulfamethoxazoles, trimethoprim, [85] sulfonamides, chloramphenicol, streptomycin, kanamycin, penicillin and gentamicin [13]	As a causative agent of chancroid endemic to Africa, Asia and Latin America [15], rates appear to be decreasing (before 2000, rates ranged from 0.0 to 69.0% geographically; after 2000, rates range from 0.0 to 15.0%) [86] except in India and Malawi [14]. Was recently identified as a causative agent of skin ulcers in children in tropical areas [87], with rates ranging from 9.0% to 60.0% [86]	Chancroid manifests as genital ulcers, in 50% of patients with genital ulcers, painful and tender inguinal lymphadenopathy may be present [13]; recently recognized to caused chronic skin ulcerations [88]	First line: ceftriaxone or azithromycin; second line: ciprofloxacin or erythromycin [14]	Gram-negative [13], despite a loose capsular structure being observed with electron microscopy [89]; *H. ducreyi* does not possess capsule-like genes, so the capsular structure produced is likely not a classical capsule [90]	Shown to require hemin and albumin [89]; studies also show media requirements differ between strains of *H. ducreyi* [91], recommended hydrolyzed protein base supplemented with complex media [89] (Mueller-Hinton, chocolatized blood agar, IsovitaleX [89,91]) at 33 °C in micro-aerophilic (increased CO2 levels) conditions for 48 h [13,91]	Genome screening of clinical isolates of *H. ducreyi* enabled to identification of some phage clusters containing predicted DNA prophages [92]. No *H. ducreyi* bacteriophages were isolated; however, other *Haemophilus* phages have been reported (*H. influenzae* phages HP1c1 [93], S2A, HP2, B, C, N3 and φflu [94] and *H. parasuis* phage SuMu [95], of which only the HP1/S2 family have been characterized in detail [94])
*Calymmatobacterium granulomatis/ Klebsiella granulomatis*	*C. granulomatis* has not been reported as an antibiotic resistance threat [17]	Endemic to specific areas of the world (India, Papua New Guinea, Brazil, South Africa [96], central Africa, northwestern Australia and the Caribbean [18]), data support a trending decrease in donovanosis over time [17]	Causes donovanosis, also known as granuloma inguinale. Infection begins with ulceration of site of inoculation, followed by lymphadenopathy. Classically there are four types of infections: ulcerogranulomatous (most common with beefy red, non-tender ulcers that bleed readily), hypertrophic or verrucous (growths with irregular edges, occasionally dry) necrotic (smelly ulcers causing deep tissue destruction) and dry, sclerotic or cicatricial lesions. Disseminated infection may occur and is usually associated with pregnancy and cervical infection [17]	Azithromycin (for a minimum of 3 weeks or until symptoms resolve) [17,97]; surgery may be required for extensive tissue damage [97]	Gram-negative intracellular encapsulated parasite of monocytes [18,97], *C. granulomatis* cells within monocyte are colloquially known as Donovan bodies [17]	Propagated within monocyte co-cultures incubated for 48 h at 37 °C in 5% CO2 [17,18]; bacteria observed intra- and extra-cellularly in monocyte co-cultures after rapid Giemsa stain [17,18]	No *C. granulomatis* bacteriophages isolated; observation of bacteriophage particles attached to and within the bacteria cell via electron microscopy has been reported, although it has also been strongly refuted [18]. A proposal exists to reclassify *C. granulomatis* as *Klebsiella granulomatis,* but there is debate based on the genetics observed [16,17]. Although no evidence supports that they may be effective against *C. granulomatis*, there are many isolated and characterized *Klebsiella* phages [40,98], with some even being used in clinical phage therapy [25,98]
*Shigella flexneri* and *Shigella sonnei*	Considered a serious threat by CDC [5], resistance to ampicillin and trimethoprim-sulfamethoxazole is nearly ubiquitous, with increasing resistance to ciprofloxacin [19], azithromycin [5] and fluoroquinolones reported [19,99]	Accounting for 5–10% of diarrheal illnesses worldwide with more than 165 million cases and 1 million deaths yearly, and despite being a gastrointestinal bacteria, *Shigella* is emerging as an STI [99], particularly among men who have sex with men (MSM). Considered an STI since 1970s [24]. Emerging epidemics in the UK of *S. flexneri* (subtype 3a—2009, 2a—2011) and *S. sonnei* (2011) among men, while rates in women have remained low [22]; epidemics suspected to target gay and bisexual men (MSM) [19]. Transmission across Europe has been observed [23]	May cause shigellosis, an acute, severe bacterial colitis [24]. Infection usually results in diarrhea (sometimes bloody), fever and abdominal pain. May cause more serious complications such as reactive arthritis [5]	Cephalosporins [21]	Gram-negative *S. sonnei* has an immunogenic O antigen group 4 capsule [100,101]	Growth observed with normal laboratory conditions (37 °C and enriched media) [102]	Many *Shigella flexneri* and *Shigella sonnei* bacteriophages have been isolated [98], characterized and sequenced (including *S. flexneri* virulent Siphophages S6 [103], pSf-2 [102] and Podophage SFPH2 [104], *S. flexneri/S. sonnei* virulent Siphophages vB SsoS-ISF002 [105] and pSf-1 [102], virulent *S. flexneri* Myophage S7 [103] and *S. flexineri*, *S. dysenteriae*, *S. sonnei* and *E. coli* C lytic Sfin-1 Siphophage [106]). Additionally, *Shigella flexneri* and *Shigella sonnei* phages have been utilized in clinical phage therapy [25,107]

## Data Availability

All Data are applicable in the paper.

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
