# Peer review of "Potential for Phages in the Treatment of Bacterial Sexually Transmitted Infections"

_antibiotics, 2021, doi:10.3390/antibiotics10091030_

Round 1

Reviewer 1 Report

This review paper is comprehensive and very well-written. The authors include current problem statements which bring to the need of implementing phage therapy and discussed about the potential challenges. Particularly in section 2 [threat of BSTIs], different aspects regarding the threats posed by different bacteria were clearly mentioned. And in section 3.4, how different bacteria affect the feasibility of phage therapy were also well-described. However, in section 3 [challenges associated with application of phage therapy in BSTI], the possible challenges were somehow lacking/vaguely presented. For example in section 3.1, the authors seem to discuss more on whether or not there are any potential phage candidate for each bacterium instead of describing about the challenges faced if we were to implement phage therapy against BSTIs. The same goes with section 3.2, which they reported mostly about the availability of phage degradative enzymes. 

There are some typos in the text, but that is not critical. Overall, this paper is very informative.

Author Response

We thank the Reviewer for her/his comments. We elaborated on possible challenges for phage and their lytic enzyme application in BSTIs in sections 3.1-3.4 according to her/his suggestions.

Reviewer 2 Report

This manuscript is well written and summarized the potential of phage therapy for bacterial sexually transmitted infections. It is good for readers to know the field. Because there are few phages available for the bacteria introduced, my only suggestion is that it will be useful to summarize the methods used for isolation of the phages in the current studies.

Author Response

We thank the Reviewer for her/his comments. We introduced a general summary on current methods of phage isolation as suggested as well we discussed some challenges in this issue.

Reviewer 3 Report

The review describes the therapeutic approach of phage against bacterial sexually transmitted infections (BSTIs). Specifically, Authors focused the manuscript on the epidemiology of BSTIs, the phages actually available (including enzymatic derivate) and clinical feasibility related to bacteria properties. Overall, the review is organized in a clear and detailed manner, so I suggest publication after minor revisions:

  • In my opinion, the review could significantly increase its impact in the clinical field by the adding of a section about the use of engineered phage. What is their opinion about it?
  • Over 65% of references are prior to 2016. More recent references are required.
  • Check out some typos, for example:
    • Sentence with singular subject and plural verb are present: "The recommended treatment for these bacteria include antibiotic" [line 27-28, page 4]; "This attributes are" (line 29, page 10)
    • after adverbs (such as Additionally [line 33, page 5]; however [line 30, page 7) the comma is generally added;
  • References should be formatted according to the style of the Journal

Author Response

We thank the Reviewer for her/his comments. Please accept a revised version of our manuscript modified according to your suggestions:

  1. we described some phage engineering approaches that may enable to produce new phages with enhanced features that could be used to treat BSTIs;
  2. we added new relevant citations (28) – most of them were published within 2019-2021
  3. citations were renumbered accordingly and formatted according to the journal style
  4. the text of the manuscript was checked by a native speaker – some grammatical errors and typos were corrected.